# Exploring Honeybee Abdominal Anatomy through Micro-CT and Novel Multi-Staining Approaches

**DOI:** 10.3390/insects13060556

**Published:** 2022-06-18

**Authors:** Jessica Carreira De Paula, Kevin Doello, Cristina Mesas, Garyfalia Kapravelou, Alberto Cornet-Gómez, Francisco José Orantes, Rosario Martínez, Fátima Linares, Jose Carlos Prados, Jesus María Porres, Antonio Osuna, Luis Miguel de Pablos

**Affiliations:** 1Grupo de Bioquímica y Parasitología Molecular CTS-183, Departamento de Parasitología, Universidad de Granada, 18071 Granada, Spain; jessicacarreira123@gmail.com (J.C.D.P.); acornetgomez@ugr.es (A.C.-G.); aosuna@ugr.es (A.O.); 2Institute of Biotechnology, Faculty of Sciences, University of Granada, 18071 Granada, Spain; 3Medical Oncology Service, Virgen de las Nieves Hospital, 18014 Granada, Spain; kevindoello@gmail.com; 4Instituto Biosanitario de Granada (ibs. GRANADA), 18014 Granada, Spain; cristinam@ugr.es (C.M.); jcprados@ugr.es (J.C.P.); 5Center of Biomedical Research (CIBM), Institute of Biopathology and Regenerative Medicine (IBIMER), University of Granada, 18100 Granada, Spain; 6Department of Physiology, Institute of Nutrition and Food Technology (INyTA), Biomedical Research Center (CIBM), Universidad de Granada, Avda del Conocimiento s/n, 18100 Granada, Spain; kapravelou@ugr.es (G.K.); rosariomz@ugr.es (R.M.); jmporres@ugr.es (J.M.P.); 7Apinevada S.L Parque Metropolitano Industrial de Granada, Calle Rubiales 17, 18130 Granada, Spain; director@apinevada.com; 8Unidad de Microscopía de Fuerza Atómica, Centro de Instrumentación Científica, Universidad de Granada, 18003 Granada, Spain; flinaor@ugr.es

**Keywords:** bee, abdomen, midgut, stain, Micro-CT

## Abstract

**Simple Summary:**

*Apis mellifera* or western honeybees are insects belonging to the Order Hymenoptera and the most important pollinators worldwide with great implications in natural biodiversity and agriculture due to their importance in pollination and honey production. The characterization of honeybee anatomy with precise tools will allow a better comprehension of the physiology of these insects under different biological conditions. Here, we employed Micro-computed tomography and novel staining methods to define the morphoanatomical characteristics of the worker honeybee abdomen. We defined the 3D and 2Ds structures of the midgut and hindgut and discovered a new cell type called ventricular telocyte, with possible roles in honeybee epithelium maintenance. Overall, we propose that this method will be useful for further investigation of the structure of the honeybee abdomen under a wide variety of environmental conditions.

**Abstract:**

Continuous improvements in morphological and histochemical analyses of *Apis mellifera* could improve our understanding of the anatomy and physiology of these insects at both the cellular and tissue level. In this work, two different approaches have been performed to add new data on the abdomen of worker bees: (i) Micro-computed tomography (Micro-CT), which allows the identification of small-scale structures (micrometers) with adequate/optimal resolution and avoids sample damage and, (ii) histochemical multi-staining with Periodic Acid-Schiff-Alcian blue, Lactophenol-Saphranin O and pentachrome staining to precisely characterize the histological structures of the midgut and hindgut. Micro-CT allowed high-resolution imaging of anatomical structures of the honeybee abdomen with particular emphasis on the proventriculus and pyloric valves, as well as the connection of the sting apparatus with the terminal abdominal ganglia. Furthermore, the histochemical analyses have allowed for the first-time description of ventricular telocytes in honeybees, a cell type located underneath the midgut epithelium characterized by thin and long cytoplasmic projections called telopodes. Overall, the analysis of these images could help the detailed anatomical description of the cryptic structures of honeybees and also the characterization of changes due to abiotic or biotic stress conditions.

## 1. Introduction

Bees are major pollinators involved in the maintenance of all terrestrial ecosystems. The western honeybee, *Apis mellifera*, is an important pollinator, initially native to Africa and expanding from Europe to western Asia around 1 million years ago [1]. These insects sustain crop pollination, the value of which is estimated at 22 billion EUR per year in Europe and 153 billion EUR per year worldwide [2]. The exact description of the internal physiology and anatomy of these insects is essential for understanding their responses to different stress factors in their ecosystem, such as pesticides, pathogens and starvation, all risk factors for bees’ survival [2].

The abdomen of the honeybee contains a number of structures comprising the circulatory, nervous, respiratory and digestive systems [3]. The digestive system occupies a relatively big proportion of the abdominal cavity and comprises the foregut (crop honey sac) and hindgut (ileum and rectum), both of ectodermal origin and the midgut (ventriculus) of endodermal origin [4]. The midgut is delimited internally by a chitinous lining of a dense matrix called the peritrophic matrix (PM) and is the main place for digestion and absorption for the honeybee [5,6]. The PM is formed of proteins, glycoproteins and chitin microfibrils organized in a proteoglycan matrix, which serves as a protective barrier of the midgut epithelium [7,8], where enzymes are secreted for the digestion and passage of nutrients to consecutive hindgut compartments [8]. The water, minerals and the rest of the products of cell metabolism excreted into the hemolymph pass to the Malpighian tubules, and later, are absorbed in the hindgut [6]. Finally, the sting apparatus located in the postabdomen is composed of the sting itself, which is formed by an unpaired dorsal component, the stylet, and a pair of ventral components, the lancets, sting sheaths, four pairs of plates, associated muscles and the accessory glands, including a poison (venom) sac [6,9,10]. The complexity of the multiple elements found in the abdominal cavity requires an accurate methodology that enables the observation of these structures at 2D and 3D levels.

X-ray microtomography (Micro-CT) is a powerful methodology used to analyze morphological and functional aspects of small animals. Micro-CT mainly consists of acquiring images in the path of an X-ray beam [11]. The source is provided by a microfocus X-ray tube that emits X-rays, which are collimated and filtered to narrow the energy spectrum. The beam then passes through the sample rotated around its long axis and is further detected with a CCD camera. The two-dimensional (2D) trans-axial projections images are collected after sample rotation at different angles. Finally, the collection of all 2D images is used for 3D stacking using specialized software [11]. This is a non-invasive non-destructive technique that allows for the visualization of samples in their original state. Micro-CT has previously been employed for the visualization of the whole body of the honeybee [12] or for the characterization of the stingers of wasps and bees [13], or for multiple brain dissections of honeybees [14] and bumblebees [15]. Moreover, other complementary non-destructive techniques, such as magnetic resonance imaging (MRI), based on interactions between protons (^1^H) in liquid phase molecules (e.g., water and lipids), have also been used to resolve drone and queen honeybees in vivo [16,17]. In this work, we analyzed, by Micro-CT, the abdominal cavity with special emphasis on the digestive tract and sting, two key elements for nutrition and defense. 

Histological analysis using hematoxylin-eosin has been widely used to describe the general morphology and histochemistry of the honeybee digestive tract. However, other staining techniques previously tested in other biological models (mice or humans) could also be useful to describe the histological compartmentalization of the honeybee gut. In this regard, Periodic Acid-Shiff-Alcian blue (PAS-AB) allows distinguishing neutral (AB+) from acid (PAS+) mucopolysaccharides. This stain has been used to characterize age-associated changes in the mucus barrier in mice colon [18], mucin distribution in paraffin-embedded colon tissues [19] or diet-induced changes in the digestive tract of Zebrafish [20]. Lactophenol stain is commonly employed to stain chitin-rich structures such as fungal cell walls [21], and Saphranin O is used as a counterstain but also to identify certain cell types such as mesenchymal stem cells [22] or chondrocytes [23]. By its side, pentachrome stain has been demonstrated as a useful technique for the simultaneous observation of multiple tissue types, such as collagen found in the extracellular matrix or sulfated mucopolysaccharides present in secretory cells [24]. Therefore, the application of these staining methods could result in a more efficient and accurate tool for the description of the digestive compartments of the honeybee.

In the present study, we have employed dual Micro-CT and Multi-staining methods to describe abdominal honeybee structures in fine detail. The results of abdominal 3D reconstruction have shown the insertion of particular structures such as the proventriculus and pyloric valve in the midgut and hindgut, respectively. Moreover, it has allowed the observation of the tissue conformation of the ventriculus and revealed a novel cell type, herein named ventricular telocyte. 

## 2. Materials and Methods

### 2.1. Honeybee Samples

*Apis mellifera iberiensis* samples were obtained from a honeybee colony located in Cortijo Tinajas (Lecrín, Granada, Spain; coordinates: 36°55′25.1″ N 3°31′42.9″ W; 36.923645, −3.528575) and kindly provided by Apinevada S. L. (https://apinevada.com/ accessed on 14 June 2022). The bees were anesthetized in the lab by placing them in close contact with a piece of cotton impregnated with a few drops of ethyl acetate for 2–5 min. The abdominal compartment was then separated using razor blades, and the gut was extracted by holding the stinger with sterilized tweezers and pulling it slowly to completely isolate the gut. For Micro-CT analysis, two honeybees were used for whole abdomen and gut analysis, whereas optical microscopy was performed with three dissected guts from three independent honeybees that were further processed and stained as described in Section 2.3.

### 2.2. Micro-CT Analysis

Sample preparation was performed by dehydrating the abdomen and gut by submerging them in subsequent 1 h incubation in increasing concentrations (70%, 90%, 100%) of ethanol solutions, Then, the samples were submerged in an ethanolic solution of iodine (1%) in 100% ethanol for 72 h. Finally, they were transferred to Hexamethyldisilazane (Sigma) overnight and left to dry at room temperature. The specimens were then kept in a dry atmosphere until a Micro-CT scan was performed. 

The abdomen and gut samples were placed and mounted on small holders. To avoid wobbling of the samples during 360° rotation, it is important that the capillary tube and pipette tip are tightly aligned along the long axis of the brass sample (Appendix A). This ensures a high-quality reconstruction. Micro-CT scans were carried out at the Centro de Instrumentación Científica (University of Granada, Granada, UGR), using a Xradia 510 VERSA (ZEISS). The following settings were established to get the same resolution of all samples: 4× magnification, 6.1515 μm voxel size, 200.021 mm source-sample distance, 20 mm detector-sample distance, BIN 1 (2048 × 2048 pixels on CCD), and 2034 projections for whole bee abdomen and 1964 for the dissected gut. The voltage, current, filter and exposure time were adjusted according to specific features of the samples as follows: bee whole abdomen: 60 kV accelerating voltage (a.v.), 83 μA beam current (b.c.), 40 s exposure time (e.t.) and LE2 source filter (s.f.); dissected gut: 40 kV accelerating voltage (a.v.), 83 μA beam current (b.c.), 60 s exposure time (e.t.) and LE2 source filter (s.f.). Image reconstruction was performed with Reconstructor Scout and Scan^TM^ (Zeiss, Oberkochen, Germany) for Center Shift and Beam Hardening Effect corrections. Dragonfly Pro^TM^ (Object Research System) was used for advanced post-processing analysis and 3D images. 

### 2.3. Tissue Staining for Optical Microscopy

Formaldehyde-fixed samples (10% formalin buffer for 48 h) of honeybee gut were included in paraffin after submerging them in consequently increasing ethanol solutions (70%, 90% and 2x absolute ethanol, 1 h each), toluene (3 × 1 h), melted paraffin (60 °C during 3 × 3 h, Sigma-Aldrich, Saint Louis, MO, USA) and finally mounting the paraffin-embedded blocks. Staining with PAS-AB (Merck) was performed to deparaffinized (xylene 2 × 10 min) and hydrated (subsequent decreasing concentrated ethanol solutions absolute, 90% and 70%, 2 × 10 min each) 0.5 μm sections [1]. All samples were stained using different staining protocols. The first one included incubation of the samples for 15 min in Alcian blue, a wash in tap water for 2 min and then in distilled water once more. Next, a 5-min incubation with Periodic Acid (in darkness) was performed, followed by a single wash with tap water. Thus far, all steps were performed at RT. Then, the samples were stained with Shiff acid in darkness and incubated for 15 min on ice. Next, samples were washed with tap water for 5 min and stained with hematoxylin for 1 min. Finally, the samples were washed again in tap water for 2 min, dehydrated in increasing ethanol solutions and xylene, following standard protocol, and finally mounted using an Organo/Limonene Mount™ (Sigma, Saint Louis, United States). The second staining protocol (L-S staining) included the incubation of the samples in Lactophenol Blue (RT, 20 min), a wash with tap water (1 min) and then staining with Safranin O (10 s). Then, the samples were washed with tap water and left to dry at RT. Finally, the samples were dehydrated in increasing ethanol solutions and xylene, following standard protocol, and finally mounted using an Organo/Limonene Mount™ (Sigma, Saint Louis, United States). The third staining protocol, pentachrome staining, was performed as previously described by Doello (2014) [2]. Hematoxylin-eosin staining was developed using the standard protocol. Briefly, samples were stained for 5 min with hematoxylin and washed in tap water for 10 min. Next, samples were stained in eosin for 5 min. Finally, the samples were washed again in tap water for 2 min, dehydrated in increasing ethanol solutions and xylene, following standard protocol, and finally mounted using an Organo/Limonene Mount™ (Sigma, Saint Louis, United States). Finally, the samples were visualized using a Nikon H600L microscope.

## 3. Results

### 3.1. Micro-CT Abdominal Reconstruction

The abdomen contains several components of the digestive, circulatory, respiratory, nervous and genital systems. The 3D reconstruction shows the digestive system (Figure 1 and Video 1) with the crop (Figure 1 region I), ventriculus (midgut) (Figure 1 region II), ileum (small intestine) (Figure 1 region III) and rectum (Figure 1 region IV). A longitudinal section shows the proventriculus invaginating into the ventriculus as a long, wide tube lying in a loop of the abdomen surrounded and connected to the Malpighian tubules (Figure 1A–D). The venom sac and the sting apparatus are visualized in the region of the terminal abdominal tergite (Figure 1B). The abdominal transverse section (Figure 1C,D) shows the digestive system, particularly the ileum, with six folds for increased nutrient absorption and a long heart placed close to the abdominal tergites.

As mentioned above, the crop and ventriculus are separated by a specialized structure called the proventriculus. According to previous reports [5], this structure can be divided into the sections (Figure 2): (I) an anterior part that is inserted into the crop lumen and forms the proventricular bulb, which consists of four lips leaving an x-shaped opening; (II) a midsection or neck; and (III) a posterior cardiac valve situated and projected to the midgut lumen. A close look at this structure (Figure 2) shows fine details of the three compartments and the anterior and posterior connection with the midgut. An example of a 2D picture (Figure 2E) shows the cardiac valve with its chitinous cuticular surface and some hair-like structures emerging on the surface towards the crop. Furthermore, this analysis shows that the valve is connected and introduced quite deeply into the ventriculus (four ventricular folds in length) with several connections between the proventriculus and the ventriculus (Figure 2C and video 2 (sec: 0:14)).

The junction between the ventriculus and the ileum is composed of a valve, known as the pylorus, where the Malpighian tubules also join. Micro-CT analysis also allowed the fine reconstruction of the pylorus on dissected worker bee guts (Figure 3). The pyloric valve is situated in the connection of the Malpighian tubules and internalized deep into the ileum, as shown in Figure 3D,E, Video 1 (Second 1 0:30 to 0:34) and Vídeo 2 (seconds 0:14 to 0:23).

### 3.2. Optical Multicolor Microscopy of Honeybee Hindgut

To analyze the different hystochemical features of the midgut and hindgut, the following staining techniques were performed: (i) Hematoxylin-Eosin (HE) for the staining of cell nuclei in purplish blue (hematoxylin) and the staining of extracellular matrix and cytoplasm in pink (eosin), with other structures taking on different shades, (ii) Periodic Acid-Schiff (PAS-AB)-Alcian blue staining for neutral and sulfated mucopolysaccharides (pink) and acid mucopolyshaccharides (light blue), (iii) Lactophenol-Saphranin for chitin staining (dark blue), iv) pentachrome staining for tissues in five fundamental colors: collagen in red; sulfated mucopolysaccharides in violet; blood cells in yellow; muscle in orange; and nuclei in green [24]. Figure 4 shows longitudinal sections of the stained ventriculus with four different methods where different regions can be differentiated (i) basal membrane, (ii) epithelium, (iii) lumen with endo- and ecto-peritrophic spaces (Figure 4). The basal membrane (i) was perfectly delimited and red stained with the pentachrome method due to the presence of collagen in this structure (Figure 4C). The columnar epithelium (ii) was positively stained for lactophenol (Figure 4D) and also showed a greenish staining plus multiple granulitic materials after pentachrome staining (Figure 4C). The lumen (iii) was extensively stained with the four methods. Neutral and mixed mucosubstances were found in the luminal surfaces of the epithelial cells and peritrophic membranes (Figure 4B). The chitin fibers were stained using the Lactophenol-Saphranin O technique, indicating the chitinous nature of this matrix (Figure 4D). The chitin fibers seem to emerge from the luminal face of epithelial cells and are completely formed as they get more immersed in the endoperithrophic matrix. The vacuolar content also emerging from the luminal face of the epithelium was strongly positive for the PAS-AB stain (Figure 4B) and pentachrome stain (Figure 4C), indicating the acidic nature of their content. All these areas show granules that were also found lining the epithelium, revealing the epithelial origin of these secretions. Within the endoperithrophic space, multiple multicolored pentachromatic digested materials resembling yeast micelia were also visualized and stained as PAS + (Figure 4E). Moreover, within the endoperithrophic matrix, multiple polymorphic structures with a basophilic circular structure resembling cell nuclei were also visualized. These structures were dispersed through the PM, resembling amoebic trophozoites (Figure 4F).

The basal membrane (i) of the columnar epithelium was tightly connected to the sub-epithelial cells, herein called ventricular telocytes. These cells have 2–3 projections per cell, from now on referred to as telopodes (Figure 5). The polymorphic (oval/piriform/spindle/triangular/stellate) cells with a positively stained morphology was slightly oval (piriform/spindle/triangular/stellate) and positively stained for Alcian blue (Figure 5D), safranin O (Figure 5C) and with the picric acid of the pentachrome stain (Figure 5B). The nucleus was stained blue with Hematoxylin (Figure 5A).

In the junction between the midgut and hindgut, the Malpighian tubules were stained with different methods, showing an empty lumen (Figure 6A–C). These slender tubules were stained green and slightly yellow after the pentachrome stain (Figure 6B), also being positive for PAS-AB staining (purple) in the close vicinity of the lumen (Figure 6A) indicating the secretory nature of urine constituents. A transversal section of the beginning of the ileum in the close vicinity of the pyloric valve showed a strong positive stain with Alcian blue, indicating the mucopolysaccharide nature of the material found in the lumen of this transversal section (Figure 6D). The ileum walls that are stained with a purple coloration indicate a mixture between PAS and Alcian blue staining and, consequently, the coincidence of neutral and acidic mucopolysaccharides in the same place (Figure 6D).

All undigested material passes from the small intestine into the rectum to be stored prior to defecation. The rectum is composed of rectal pads, which are in charge of reabsorbing minerals and water. These structures were positive for Alcian blue, indicating the presence of acid mucins in the cells (Figure 7 and Appendix A). Inside the rectal pads, the nucleus of columnar cells was stained with hematoxylin-eosin (Appendix A) and the junctions between those cells were stained by hematoxylin-eosin, pentachrome and PAS-Alcian blue stains. The positivity for lactophenol blue of the supra epithelial lining indicates the presence of chitin, which is delimiting the rectal pads (no significant results were found for Lactophenol-Saphranin stain) (Appendix A).

## 4. Discussion

Since the abdomen of honeybees is involved in numerous physiological processes, the discovery of novel structures/markers and how they change could respond to different biotic or abiotic stressors during the lifespan of a honeybee is of great relevance. In this sense, Micro-CT and novel staining methods have led to the discovery of cryptic structures that otherwise would not be appropriately characterized.

Here, the Micro-CT technique allowed the detailed characterization of the proventriculus and pyloric valves that connect the foregut, midgut and hindgut. Previous reports have characterized the proventricular structure with its x-shaped proventricular bulb and the sclerotic cuticular intima by scanning electron microscopy and Micro-CT [25,26]. In our work, an extensive elongated projection of this valve into the ventricular lumen that radiates all the way through the lumen of this structure was found. At this location, the proventriculus seems to be prolonged towards the midgut merging with nervations and prolongations (Figure 7). Although the chitinous nature of the peritrophic membrane could also be the source of these projections, these data show a longer connection of the cardiac valve and its continuation into the midgut to what was described to date. By its side, studies on the pyloric valve are quite scarce [9,27]. Some authors have proposed that the pyloric valve prevents the reflux of the hindgut content into the midgut [27]. The Micro-CT data is in accordance with the transmission electron microscopy study performed by Serrão and Cruz Landim (1996) [26], which indicates that the valve is inserted into the lumen of the midgut right after the junction with the Malpighian tubules. Moreover, 3D and 2D Micro-CT scans showed the occlusion of the pyloric valve from the midgut to the hindgut, which is also in accordance with previous descriptions of this valve, ensuring the proper compartmentalization of the midgut and hindgut [27,28,29]. Moreover, this work has found the positive staining of the lumen of the pyloric valve for Alcian blue, indicating the presence of acid mucins and hence active secretions as suggested by the presence of proliferating epithelial cells at this side (Figure 7).

One of the most striking results of the histomorphological analysis was the discovery of a series of pericryptal sub-epithelial polymorphic cells underneath the ventricular epithelial invaginations, defined herein as ventricular telocytes. These cells have been previously described in other multicellular models [30,31,32] and defined as part of the stem-cell niche [33], implicated in tissue regeneration as they provide a series of growth factors that maintain stem and progenitor cell proliferation in different organs such as the intestine, skeletal muscle, heart, lung or skin [34]. Intestinal telocytes express the winged-helix transcription factor Foxl1 and the hedgehog signaling mediator Gli1, both acting as growth factors [33]. Ventricular telocytes’ characteristics are in accordance with the definition of this cell type, with their basal lamina occasionally being present and in contact with smooth muscle cells. They usually have 2–3 cytoplasmic projections, ranging from several tens to hundreds of micrometers in length, forming three-dimensional networks [35]. Moreover, ventricular telocytes of honeybee were positively stained with Alcian blue, which indicates the presence of acid mucins or glycosaminoglycans, in accordance with recent reports in fish telocytes [36]. As described in other multicellular organisms [34], these cells could be implicated in tissue regeneration, especially since epithelial cells are subjected to daily potential aggressors (i.e., pathogens, abrasive food particles from foraging of nectar or pollen).

The histological staining and the use of novel staining methods for the peritrophic membrane is also an adequate tool to describe honeybee diet components as well as ecto-endo peritrophic matrix morphology or disposition. The detection of acidic vacuolar material, by optical microscopy, accumulated in the peritrophic membrane, indicates the possible presence of proteolitic enzymes in granules that emerge from the epithelial cells leading to the endoperithropic space. In this sense, pentachrome staining has proven a good marker for labeling diet structures engulfed by the peritrophic membrane since it has strongly acidic compartments. These results are supported by the positive staining for PAS-AB of the same endoperithrophic space. The PAS stain has been previously used for staining of Nosema ceranae spores allowing the observation and quantification of those pathogens in the midgut of the honeybee thus the combination of pentachrome and PAS staining could also serve to describe either pathogens or components of the bee diet [37]. In this sense, we have noted the presence of what resembles cells of a non-defined nature in the peritrophic matrix, observed after the four staining procedures. The ameboid morphological status and the unique basophilic circular structure of these potential cells could indicate the presence of symbiotic or pathogenic amoeba throphozoites. In this regard, Malpighamoeba mellificae have been previously identified as parasitic protozoan amoeba invading the Malpighian tubules as cyst forms [38]. There are no previous reports on the presence of the vegetative trophozoite forms of this parasite in the ventricular space. Therefore, further research to verify the presence of such organisms in honeybee midgut would be of interest.

## 5. Conclusions

In this work, Micro-CT and multiple staining methods have resolved several features of the worker honeybee abdomen. Here, the precise structure and connections of proventricular and pyloric valves, including the extended length of the posterior proventricular cardiac as well as the 3D structure and the acidic nature of the muchopolysaccarides located within the pyloric valve were detailed. Moreover, multi-staining approaches have allowed the location of ventricular telocytes in honeybees to be found for the first time. Given the essential role in tissue remodeling and homeostasis in other multicellular organisms, further research on this particular cell type would be of special interest for the understanding of midgut epithelium dynamics and maintenance. The results presented here are fully accessible and can provide a solid background for further investigations to discover functional and morphological comparisons among different bee species or conditions.

## Figures and Tables

**Figure 1 insects-13-00556-f001:**
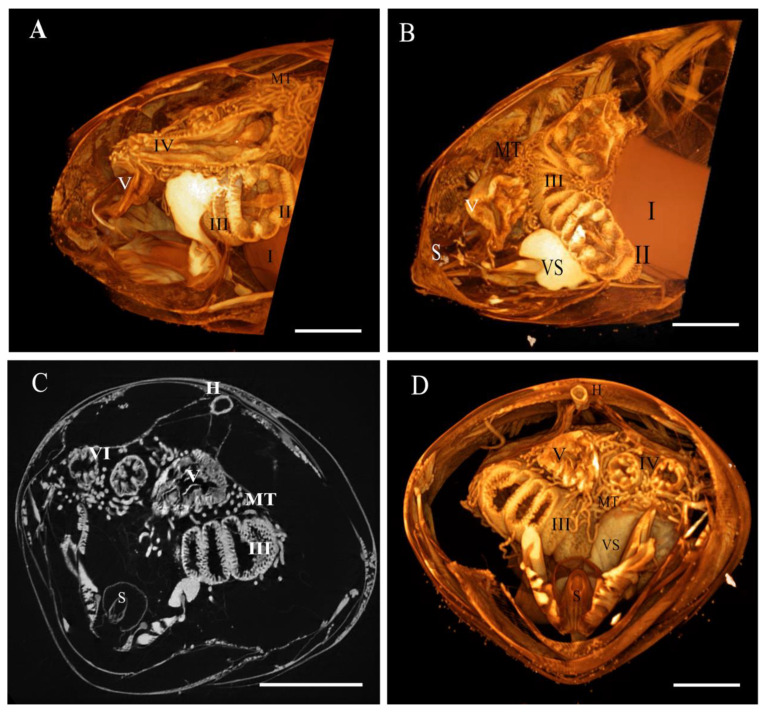
Tomography of the worker honeybee abdomen. (**A**,**B**) Images of the abdomen in a longitudinal direction; (**C**) 2D transverse section of the abdomen; (**D**) 3D transverse section of the abdomen. (I) Crop, (II) proventriculus, (III) ventriculus, (IV) ileum, (V) rectum, (MT) Malpighian tubules, (VS) venom sac, (S) sting and (H) heart. Bars: 1 mm.

**Figure 2 insects-13-00556-f002:**
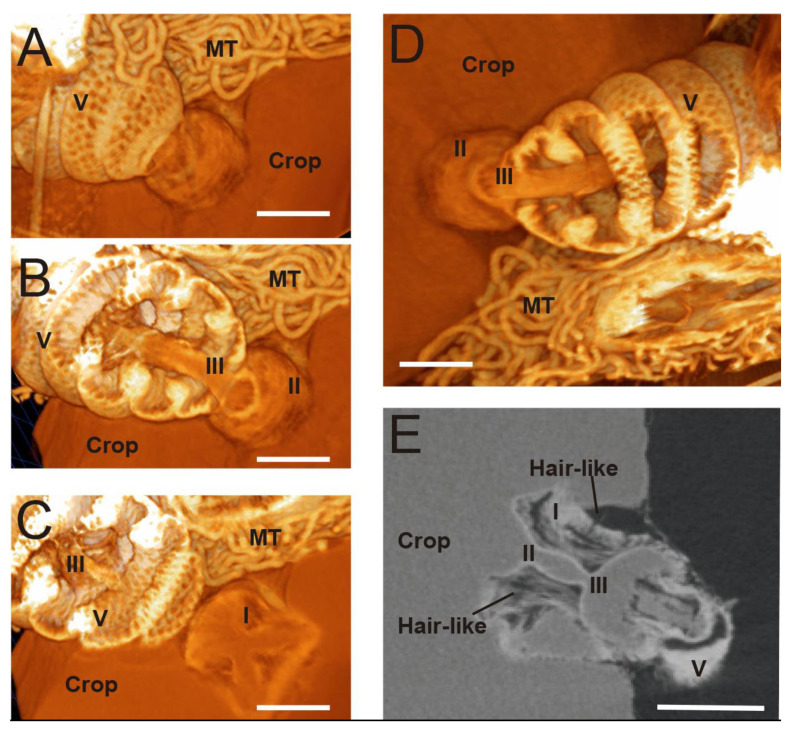
Tomography images of the proventriculus structure; (**A**) Connection among honeybee crop-proventriculus-ventriculus; (**B**) Section showing the neck (II) and protrusion into the lumen of the midgut (III); (**C**) The x-shape folds on the proventriculus protrusions (I) and projects into the lumen of the midgut (III); (**D**) More detailed image of the proventriculus cardiac valve inserted in the midgut lumen showing the neck (I) and posterior part (III); (E) 2D scan with anterior part (I), showing the neck (II) and posterior part (III), internalizing into the lumen of the ventriculus (V). V: ventriculus, MT: Malpighian tubules. Scale bar: 250 µM.

**Figure 3 insects-13-00556-f003:**
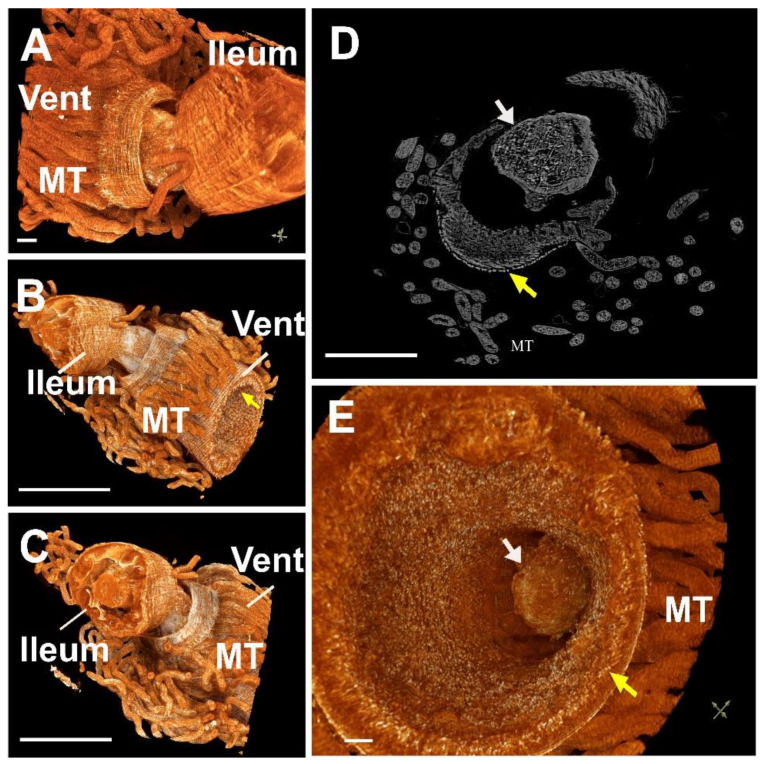
Tomography images of the connection between the midgut and the hindgut through a pyloric valve. (**A**) The ventriculus with Malpighian tubules (MT) and peritrophic membrane (yellow arrows); (**B**,**C**) Ventriculus with highlighted ectoperitrophic space; (**D**) 2D image; (**E**) Highlighted ectoperitrophic space. Bars: (**A**,**D**,**E**) = 100 μM; (**B**,**C**) = 1 mm.

**Figure 4 insects-13-00556-f004:**
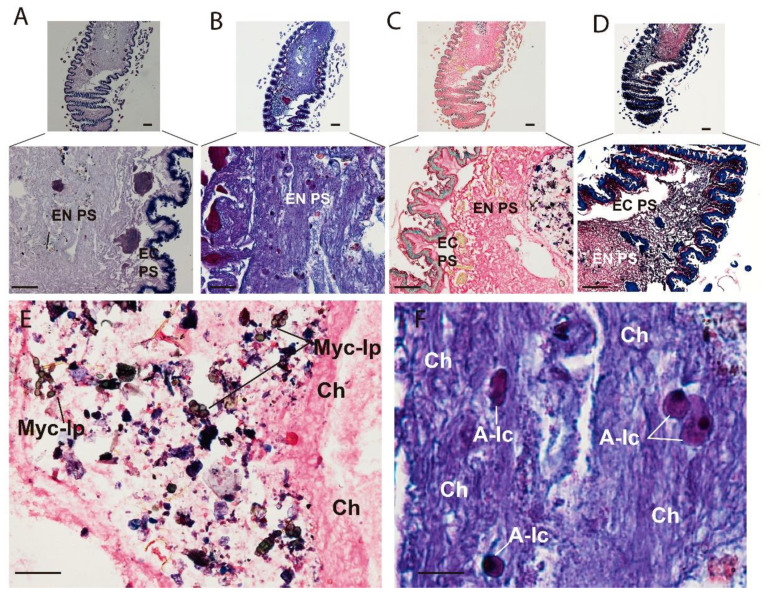
Multicolor stain of ventriculus reveals multiple components of the perithrophic membrane. (**A**) Hematoxylin-eosin stain; (**B**) PAS-AB stain; (**C**) Pentachrome stain (yellow structures are stained due to picric acid); (**D**) L-S stain; (**E**) Higher magnification of the food bolus stained with pentachrome stain; (**F**) Higher magnification of nucleated cells within the peritrophic membrane stained with PAS-AB. EN PS: Endoperithropic space; EC PS: Ectoperithrophic space; Myc-lp: Mycelium-like particles; A-lp: Ameboid-like cells; Ch = Chitin. Scale bar = 100 nm.

**Figure 5 insects-13-00556-f005:**
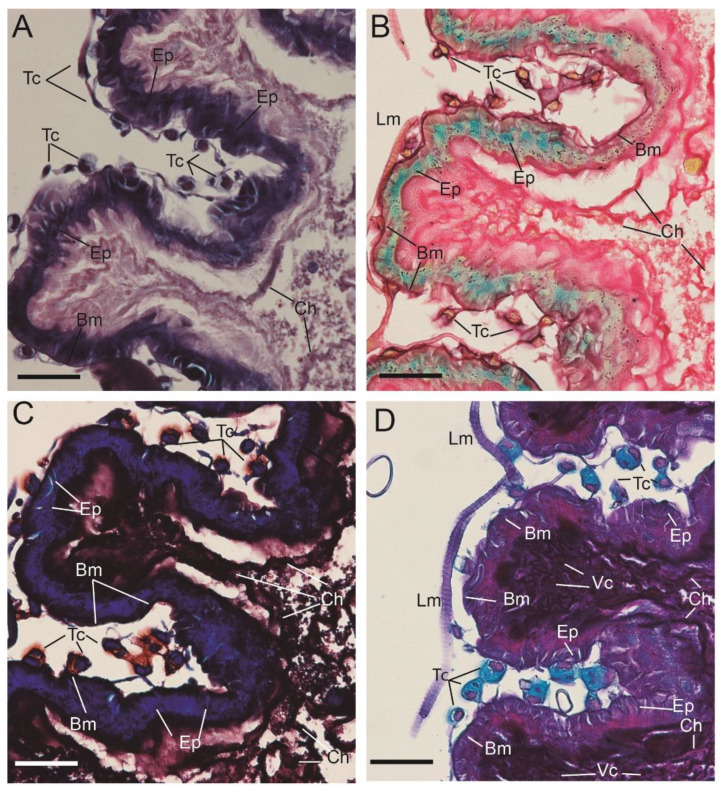
The ventricular epithelium is surrounded by an extensive line of telocytes. (**A**) Hematoxylin-eosin stain; (**B**) Pentachrome stain; (**C**) Lactophenol-Safranin stain; (**D**) PAS-AB stain. Lm: Longitudinal muscle; Tc: telocyte; Bm: Basal membrane; Ep: Epithelium, Ch: Chitin. Scale bar = 100 nm.

**Figure 6 insects-13-00556-f006:**
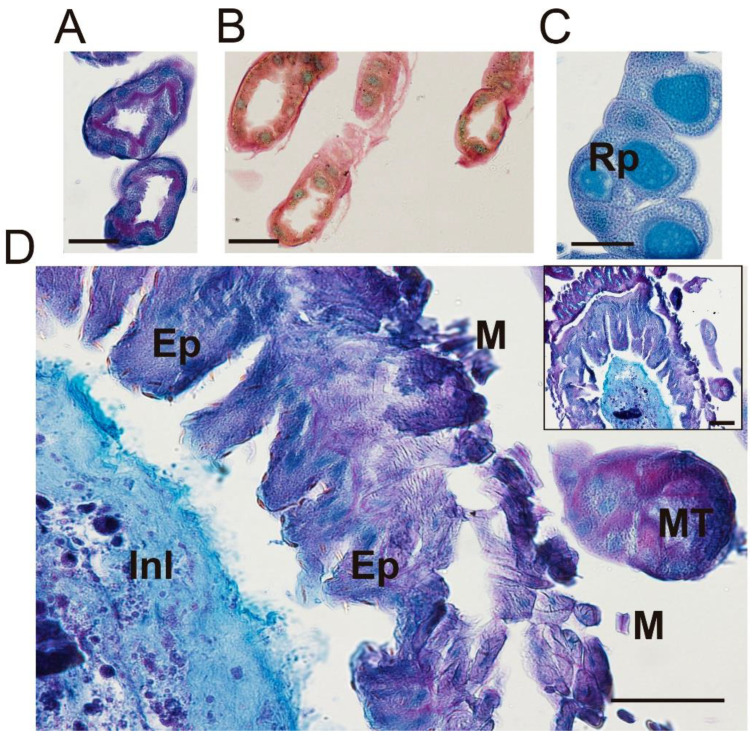
Optical microscopy of Malpighian tubules, rectal pads and the pylorus. (**A**) Malpighian tubules stained with PAS-AB. Scale bar = 50 nm; (**B**) Malpighian tubules stained with pentachrome staining. Scale bar = 50 nm; (**C**) Rectal pads stained with PAS-AB. Scale bar = 100 nm; (**D**) Alcian blue-positive dense material in the lumen of the pyloric region. Rp: Rectal pad; Ep: Epithelium; Inl: Inner lumen; MT: Malpighian tubules; M: Muscle layer. Scale bar = 100 nm.

**Figure 7 insects-13-00556-f007:**
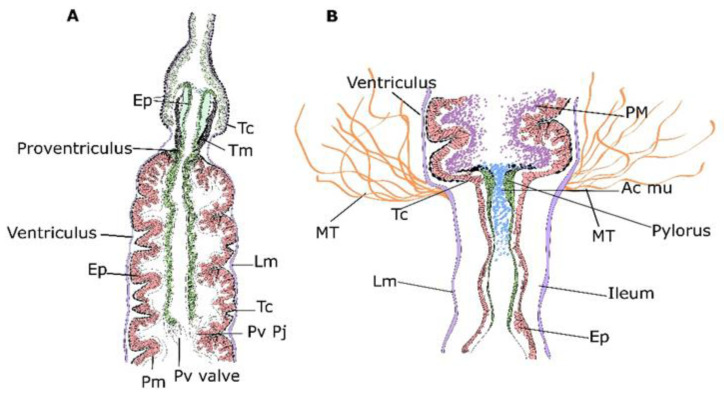
Diagrammatic representation of the junctions between the foregut, midgut and hindgut. (**A**) Longitudinal section of the proventriculus and the ventriculus. (**B**) Longitudinal section of pylorus and ileum. Epithelium (Ep), telocytes (Tc), longitudinal muscles (Lm), peritrophic membrane (Pm), proventricular valve (Pv valve) and proventricular projections (Pv Pj), Malpighian tubules (Mt) and Acid mucopolysaccharides (Ac mu).

## Data Availability

The 2D scans of honeybee abdomens are available at Zenodo (https://zenodo.org/record/6571775 accessed on 14 June 2022) under the Digital Object Identifier 10.5281/zenodo.6571775; The Video 1 is available at Zenodo (https://zenodo.org/record/6390165, accessed on 14 June 2022) under the Digital Object Identifier: 10.5281/zenodo.6390165; The Video 2 is available at Zenodo (https://zenodo.org/record/6657838, accessed on 14 June 2022) under the Digital Object Identifier: 10.5281/zenodo.6657838.

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
