# Peer review of "Exploring Honeybee Abdominal Anatomy through Micro-CT and Novel Multi-Staining Approaches"

_insects, 2022, doi:10.3390/insects13060556_

Round 1
Reviewer 1 Report
Thanks a lot for the revision, I think most of the points have been clarified. Just one question: Would it be possible to have a scale bar as well in the 3D scans? Or any "rough" indication how big things are?
Author Response
Review Report 1
1.Thanks a lot for the revision, I think most of the points have been clarified. Just one question: Would it be possible to have a scale bar as well in the 3D scans? Or any "rough" indication how big things are?
Response: In order to provide an approximation of “how big things are '' we have added a scale bar, which is possible using Dragonfly ™ (Zeiss) software for Micro-CT imaging. However we should puntualize that since we are imaging 3D volumes this is just an approximation since structures at the front or the sides have different depths. In anycase, we appreciate the referee's comments since now the reader has an idea of the scale of the structures indicated on the figures.
Reviewer 2 Report
Dear Authors,
The manuscript aims to improve our knowledge about the abdominal morphology of the honeybee using micro-CT and histochemical methods. As the model organism is of great importance, the manuscript is expected to attract the attention of a wide readership. The most exciting result of this research is the discovery of ventricular telocytes in honeybees which is a specific type of cells associated with the midgut epithelium. The methodological innovation of the study is employing multi-staining methods as a more efficient and accurate tool for describing the digestive tract in the honeybee.
There are some weak points of the manuscript to review mostly related to the morphological descriptions and interpretations.
Please see specific comments in the attachment.
Good luck!

Author Response
A brief summary
The manuscript aims to improve our knowledge about the abdominal morphology of the honeybee
using micro-CT and histochemical methods. As the model organism is of great importance, the
manuscript is expected to attract the attention of a wide readership. The most exciting result of this
research is the discovery of ventricular telocytes in honeybees which is a specific type of cells
associated with the midgut epithelium. The methodological innovation of the study is employing
multi-staining methods as a more efficient and accurate tool for describing the digestive tract in the
honeybee.
General concept comments
There are some weak points of the manuscript:
- the scientific novelty of the micro-CT results is not clear, at least, at this stage of the
manuscript preparation (see Specific comments);
- some of the morphological terms require reviewing and unification throughout the text;
more structures have to be indicated on pictures, especially Figs 5 and 7, Suppl. Fig1;
Response: We really appreciate the comments since they have enriched the manuscript. We have added the terms required by the referee (see specific comments on the figure 5, 7 and Supplementary Fig. 1 below). We have also carefully checked for the uniformity of the terms throughout the text so now it must be uniform throughout the manuscript.
- the main text suffers of lacking descriptions of tissues and structures shown on pictures,
authors have to use a classic work of Snodgrass (1910, The anatomy of the honey bee) to
discuss the structures in a comparative aspect;
Response: We have carefully read the work of Snodgrass (1910) and discussed in the text. Specially the part regarding the proventricular and pyloric valves. Indeed we have added Fig 45 (Snodgrass) reconstruction adapted the original indicating the novelties found in the proventricular-ventricular joint by Micro CT (Figure 2).
- a generalized picture showing relative position of target structure inside the abdomen may help to understand the descriptions: it can be a semi-diagrammatic drawing or micro-CT volume rendering like this one https://bit.ly/3yXkKR7 (doi: 10.3390/insects13020185) and this one https://www.nature.com/articles/s41598-019-53537-z/figures/2
Response: We really appreciate the advice and tips provided by the referee that have enriched our explanations on the structures imaged by Micro-CT. We have introduced a diagrammatic drawing on figure 2 and Figure 3 indicating the findings of our study. In the case of Figure 2 this figure is an adaptation of the original Figure 45 of Snodgrass (1910) as commented above.
Specific comments
Please correct this throughout the main text and in the figure captions:
- honey sac > crop;
Response: Done
- please avoid using of term ‘stomach” in a honey bee. You can find the honey stomach (=
crop) and true stomach or “chyle stomach” (=ventriculus) in the publications. Therefore,
“crop” and “ventriculus” are preferred terms;
Response: Done
- malpighian tubules, Malpighian tubes > Malpighian tubules;
Response: Done
- stinging apparatus > sting apparatus;
Response: Done
- Telocytes > telocytes
Response: Done
- use either “basal membrane” or “basement membrane”, but don’t mix two terms;
Response:Done. We have replaced “basement membrane” by “basal membrane”
- We have included the following paragraph at the end of the section 2.3 Tissue staining for optical microscopy: “Finally, the samples were visualized using Nikon H600L microscope”.
P2, L60 “is composed of” > “contains” (because the abdomen composed of a number of structures including skeletal elements, trachea ect.)
Response: Done
P2, L62 “big proportion on the honeybee abdomen” > ” big proportion on the abdominal cavity”
Response: Done
P2, L62 “(crop)” > “(crop, honey sac)”
Response: Done
P2, L64 “endodermic” > ”endodermal”
Response: Done
P2, L64 “is delimited by” > “is delimited internally by”
Response: Done
P2, L71-72 “the last abdominal segment” > “the postabdomen” (because the sting chamber is
formed by an infolding of 8th, 9th and 10th segments into the 7th segment, indeed)
Response: Done
P2, L72-75 “by the sting which is formed….and a bulb” > “the sting itself which is formed by an
unpaired dorsal component the stylet and a pair of ventral components the lancets, sting
sheaths, four pairs of plates, associated muscles, the accessory glands, including a
poison (venom) sac” (please note, that the bulb is a part of the stylet, so it is not a
separate structure)
Response: Done (lines 73-75)
P2, L79-83 Three sentences “It mainly consists of acquiring images… or drawing.” can be
excluded since they describe a routine procedure of micro-CT analysis, not related
directly to the topic of this MS.
Response: Done
P3, L110 “insertion… into the abdominal cavities” > “intact position…in the abdominal cavity”
Response: Done
P3, L118 “sacrificed” - killed? is that correct verb?
Response: We have changed the term “sacrificed” by “anesthetized”.
P3, L138 “(Xradia 510 VERSA (ZEISS)” – delete repetition
Response: Done
P4, L156-166 “at RT” > “at room temperature”?
Response: Thank your observation. The “RT” was replaced by “room temperature”
P4, L161 “with L-S” > “with Lactophenol-Saphranin”?
Response: We replaced “L-S” by “Lactophenol-saphranin”
P4, L161 “with Lactophenol Blue” > “with Lactophenol Cotton Blue”?
Response: We replaced “Lactophenol Blue” by “Lactophenol Cotton Blue”
P2, L170-172 “The worker honeybee abdomen is a large structure composed by the circulatory,
respiratory and digestive systems, abdominal muscles and nervous system composed by
5 ganglia and the sting apparatus” > “abdomen contains several components of digestive,
circulatory, respiratory, nervous and genital systems”
Response: Done (lines 187 and 188)
P4, L173 “honey sac” > “crop”
Response: Done (and also change this term in Figure 1).
P4, L174 “A longitudinal section shows the proventriculus invaginating into the ventriculus, as a
long, wide tube, lying in a loop of the abdomen surrounded and connected to the
malpighian tubes (Figure 1A-D).” – please indicate the proventriculus on the figure 1
Response: Thank you for your observation. We adjusted the figure 1 and rewrote the subtitle of the figure 1A-D (line 198-201).
P4, L176-177 “malpighian tubes” > “Malpighian tubules”
Response: Done
P4, L177-178 “Next to the rectum, the venom sac and the sting are placed in the terminal
abdominal tergite” > morphologically more correct is to write: “The venom sac and the
sting apparatus are visualized in the region of the terminal abdominal tergite”.
Response: Done (lines 187 and 188).
P4, L180-181 “Figure 1C-D also provides a close look to the sting plates and abdominal muscles.”
– I don’t see clearly neither sting plates or separate muscles on this figure. The figure 4 is
much more informative. Either delete this sentence or provide reference on fig4
Response: Thank you for your observation. Since we removed Figure 4 from the text (micro-ct imaging of the 6th ganglia and TAG and its connection with the stinging apparatus) we decided to also delete the sentence “Figure 1C-D also provides a close look to the sting plates and abdominal muscles.”
P5, L187, 189 “honey sac” > “crop”
Response: Done
P5, L188 “valve called…” > “structure called…” (to avoid misunderstanding, because the third
region of the proventriculus is the cardiac valve)
Response: Done
P5, L195 “some segmented hair emerging on the surface towards the crop” - pls indicate these
hair (hairs?) on the figure you mentioned.
Response: The hair-like structures are now indicated in Figure 2E.
P5, L197 “with several connections emerging from the ventriculus towards the proventriculus” >
pls indicate these connections on the figure you mentioned.
Reponse: We appreciated this comment too. Indeed, after a careful observation of the results we concluded that proventriculus is continued by those connections thus we have changed the sentence “with several connections emerging from the ventriculus towards the proventriculus” by “several connections between proventriculus and the ventriculus (Figure 2C and video 2 (sec :0:14)).” (lines 208-213). For more detail we have indicated these projections in Figure 2C and in second 0:14 of Video 2 (lines 208-213)..
P6, L210 “malpighian tubules” > “Malpighian tubules”
Response: Done
P6, L211 “proventriculus” > “ventriculus”?
Response: Done
P6, L211 “may increase the flexibility of this structure” – can you please explain this idea about
flexibility?
Response: We appreciate the referee's comments and thought about the observations made in Figure 3. In this figure we have found that the external tissue found between the connection of the ventriculus and ileon is not a continuum, having a disconnection that may generate a space for both structures to bend and be more flexible. Given that our analysis is based in one individual we could not conclude that this is not an artifact created by the force generated by pulling on the digestive tract during the dissection procedures.Therefore, we eliminated this sentence from the text. We will definitely need to confirm this observation by increasing the number of guts analyzed by Micro-CT in future research.
P7, L218-224 This part of the text related to the sting apparatus seems to be not really informative
because of two reasons: (1) the fragmental and not complete descriptions – both in the text and in
figures; (2) the absence of comprehensive discussion of the sting structures observed next in the
Discussion part. I realize that authors aim is to introduce a structural background to the innervation
briefly discussed later. However, the nerve roots running from abdominal ganglia to inner
abdominal structures still are insufficiently and questionable described and visualized (see also
below). For example, there is a number of structures to observe on Fig4, but only 6th ganglion,
TAG and some nerve roots are highlighted and mentioned in the text. I don’t see what exactly
structures are innervated by specific roots. So, either delete this part of the Results or provide detailed and morphologically correct descriptions (see Snodgrass 1910: https://naldc.nal.usda.gov/download/CAT31027153/PDF , more recent Packer 2003: https://doi.org/10.1046/j.1096-3642.2003.00055.x or Stetsun & Matushkina 2020: doi: 10.1111/ens.12438).
Response: We really appreciated the referee's comments. In the previous version of the manuscript we illustrated the Terminal Abdominal Ganglia (TAG) and the 6th ganglia as structures that have never been explored by Micro-CT and in such detail before. However, the more zooming the lower the resolution we obtained in our Micro-CT scans. In our previous response (see Figure 4 of our previous MS version) we isolated the nerviations and TAG and 6th and tried to provide a detailed view of the ganglia and nerve fibers. Although we successfully resolved the different nerves emerging from the 6th and TAG, with our Micro-CT resolution in hand, we could not precisely describe the connections between the nerve roots with the stinging apparatus as required by the referee 2. Therefore we decided to finally remove Figure 4 so this way we would avoid any possible misinterpretation or mistake in the description of sting apparatus connections with 6th and/or TAG.
P9, L239 abbreviations LS and PS can be deleted as they are not used in the next text.
Response: Done
P9, L240-241 “red blood cells in yellow” - as red blood cells absent in bees, can you explain what
exactly component of the cell can be stained in yellow in the honey bee?
Response: Thank you very much for your appreciation. Picric acid stains the cytoplasm of red blood cells in yellow because this compound stains basic proteins. Therefore, bee cells (in their cytoplasms)or mucosubstances that contain basic proteins would be stained in yellow.
P9, L243-244 “lumen with endo- and ecto- peritrophic spaces (Figure 5)” – indicate the
compartments of the gut + peritrophic membrane in Figure 5; probably refer to Fig6 too.
Response: Thank you very much for the suggestion. In order to clarify we have indicated where the endo- and ecto-peritrophic spaces are located in the images provided in Fig. 4. We believe that the indications on the figure should facilitate the visualization and location of these structures to the reader.
P9, L262, 267 PM > peritrophic membrane?
Response: We changed “peritrophic matrix” and “PM” by “peritrophic membrane”.
P10, L270 “ventricular Telocytes” > “ventricular telocytes”
Response: Done
P10, L271 “(Tp)” – please delete as this abbreviation is not used neither in the text
Response: Done
P10, L271-272 “Their morphology was slightly oval (piriform/spindle/triangular/stellate) and
positively stained…” > “Polymorphic (oval/piriform/spindle/triangular/stellate) cells were positively
stained…”
Response: Done
P10, L279 “malpighian tubules” > “Malpighian tubules”
Response: Done
P10, L280 “Slender tubes” > Malpighian tubule or some specific region of the tubule?
Response: We appreciated the referee's comments. We referred to Malpighian tubules. To avoid any confusion we changed the previous sentence “Slender tubules were stained green and slightly yellow after pentachrome stain” by “These slender tubules were stained green and slightly yellow after pentachrome stain”s” (lines 285 and 286).
P10, L283-P11, L286 “A transversal section of the beginning of the ileum in the close vicinity of the
pyloric valve showed a strong positive stain with Alcian blue indicating the mucopolysaccharide
nature of the material found in the lumen of this transversal section” > “A transversal section of the
beginning of the ileum in the close vicinity of the pyloric valve showed a strong positive stain with
Alcian blue indicating the mucopolysaccharide nature of the material found in the lumen”. What
about the ileum wall? Can you describe it?
Response: Thank you very much for the comments. The ileum wall presents a predominance of acid mucopolysaccharides, so there is a light blue staining there (alcian blue positivity). Moreover, there is a small PAS positivity (neutral mucopolysaccharides, pink) in the deepest regions of the ileum wall but much smaller than the ventricle. Therefore, the regions of the ileum wall that are stained with a purple coloration indicate a mixture between PAS and alcian blue staining and, consequently, the coincidence of neutral and acidic mucopolysaccharides in the same place. Accordingly and to enhance the precise description, we have added the following sentence in the text: “The ileum wall that are stained with a purple coloration indicating a mixture between PAS and alcian blue staining and, consequently, the coincidence of neutral and acidic mucopolysaccharides in the same place (Figure 6D)” (lines 291-294).
P11, L294-296 Please provide description of tissues and structures visualized on pictures with
different staining methods. Probably, you can use some text from captions of Supplementary
Figure 1.
Response: We appreciate observations provided by the referee. For a better description of the rectum we have modified the previous paragraph: “These structures were positive for alcian blue indicating the presence of acid mucins (Figure 7C). They were positively stained in red and dark blue after Pentachrome and PAS-AB staining, respectively (no significant results were found for Lactophenol-Saphranin stain) (Supplementary Figure 1)” by the following one “ These structures were positive for alcian blue indicating the presence of acid mucins in the cells (Figure 7C and Supplementary Figure 1). Inside rectal pad,the nucleus of columnar cells was stained with hematoxilin-eosin (Supplementary Figure 1B) and the junctions between those cells stained by hematoxilin-eosin, pentachrome and PAS-alcian blue stains. The positivity for lactophenol blue of the supra epithelial lining, indicates the presence of chitin which is delimiting rectal pads (no significant results were found for Lactophenol-Saphranin stain) (Supplementary Figure 1)” (lines303-310).
P12, L315 “by scanning electron microscopy and Micro-CT… In our work, an extensive elongated
projection of this valve into the ventricular lumen that radiates all the way through the lumen of this
structure was found… Although the chitinous nature of the PM could also be the source of these
projections this data shows a longer connection of the cardiac valve and its continuation into the
midgut to what was described to date” – What about Fig 45 (P97) of Snodgrass 1910:
https://naldc.nal.usda.gov/download/CAT31027153/PDF ? Can you please compare your data with
the data of Snodgrass, if applicable? Why do you think that the elongation you found on micro-CT
scans is not a peritrophic membrane indeed? How does the micro-CT compare to histology?
Response: These are all excellent suggestions, and appreciate comments particularly the references provided by the referee. After carefully reading the work of Snodgrass (1910, Figure 45), we agree that the author have previously described proventricular tubular folds opening into the the ventriculus as described in the following paragraph (Snodgrass 1910): “The posterior opening of the proventriculus into the ventriculus is guarded by a long tubular fold of its epithelium (fig. 45, PventVlv), the proventricular valve. This would appear to constitute an effective check against the escape of any food back into the proventriculus. It looks like one of those traps which induces an animal to enter by a tapering funnel but whose exit is so small that the captive can not find it from the other side.”.
Since the Micro CT provides a clear 3D image of the “long tubular fold” described by Snodgrass we confirmed the illustrations provided in Figure 45 by Snodgrass. The Micro-CT analysis shows that the valve is connected and introduced quite deeply into the ventriculus (4 ventricular folds in length) with several connections between proventriculus and the ventriculus (Figure 2C and video 2 (sec :0:14)).
Thus, we have adapted the Figure 45 of Snodgrass work (1910) to describe the morphology of proventriculus and its connection with the ventriculus in a new figure (Figure 7) summarizing the findings including the extensión of the long tubular structure of the posterior proventriculus up to the 4th ventricular fold and the telocytes underneath the epithelium layer.
Regarding whether or not the elongations found correspond to the Peritrophic Matrix, there is definitely the possibility that some of the nerviations and elongated structures within the lumen of the ventriculus corresponds to the peritrophic matrix. Indeed in minute 1:01 of the video 1 and 0:12 of video 2 there is a clear picture of what it seems to be a peritrophic matrix especially due to the irregular nature of the nerviations and projections within the lumen of the ventriculus. We have indicated the presence of those projections in the diagrammatic picture illustrated in figure 7A.
P12, L319; P13, Lines 366,369, 372 PM > peritrophic membrane
Response: Done
P12, L335-P13, L347 and also P14, L388-390 I am afraid, the micro-CT approach alone does not
provide a clear picture of innervation because of its roughness. There is a number of structures
located in the post abdomen including skeletal musculature, trachea and several sclerites with
sensilla, so how can you be sure that the nerve roots you observed run to any sting structure? And
if so, what specific structure is innervated? Moreover, the sting apparatus with components
described in the Introduction originates from 8th and 9th abdominal segments, so its innervation by
the 6th ganglion is very unlikely. If I am not right, provide more details and compare with available
literature exploring innervation with more traditional methods.
Response: As commented above, we have evaluated the possibility of describing the structures connected with the 6th and TAG ganglions. The limits of resolution provided by our Micro-CT analysis do not allow the precise description of the structures innervated with the risk of potential mistakes on the morphological identification. Therefore and in spite of the novelty in the 3D-visualization of the TAG and 6th ganglia we have finally decided to eliminate Figure 4 “Tomography images of the connection between the stinging apparatus and the abdominal ganglia.” from the MS to avoid any misconfusion in the identification of innervated structures. We have also removed the references to the sting apparatus throughout the text.
P13, L350 “pericryptal sub-eptihelial elongated cells” > “pericryptal sub-eptihelial polymorphic
cells”. If there is a pericryptal region, please indicate “crypts” on Figures and in the text of the
Results.
Response: We appreciate referees suggestions although we respectfully believe that the term perycryptal indicates with precision the location of telocytes. However we modified the MS text and included “underneath of the ventricular epithelial invaginations” to complete the following sentence: One of the most striking results of the histomorphological analysis was the discovery of a series of pericryptal sub-epithelial polymorphic cells underneath of the ventricular epithelial invaginations defined herein as ventricular telocytes.”
P13, L360 “ventricular telocytes were” > “ventricular telocytes of honeybee were”
Response: Done
P13, L374 “N. ceranae” > “Nosema ceranae”
Response: Done
P13, L375 “in the midgut, thus” > “in the midgut of honeybee, thus”
Response: Done
P14, L390 “stinging apparatus” > “sting apparatus”
Response: We have removed the sting apparatus study from the text.
FIGURES
please don’t use the abbreviated names of the staining methods in captions, since not all readers
are familiar with such abbreviations.
Figure 1
Check: A,B – longitudinal sections, C,D – transverse sections. Bulb of the sting is a part of sting,
see comments above. Indicate direction on the pictures.
Response: We have included referee´s indications in figure 1 legend that we attached here: Figure 1. Tomography of the worker honeybee abdomen. In (A) and (B) images of abdomen in longitudinal section; (C) 2D transverse section of the abdomen; (D) 3D transverse section of the abdomen (I) crop, (II) proventriculus, (III) ventriculus, (IV) ileum, (V) rectum, (MT) Malpighian tubules, (VS) venom sac, (S) sting and, (H) heart. Bars: 1 mm (lines 198-201).
Figure 2
honey sac > crop
there are two ”V” on Fig2D – check please
Response: We have made modifications required on figure 2.
P6, L204 – “showing the neck (I)” > “showing the neck (II)”
Response: Done
Figure 3
Indicate ventriculus, ileum, and pyloric valve, if applicable.
Response: Done
Figure 4
Please see my comments on the sting apparatus issue. If you decide to develop the description in the main text, please indicate all the possible structures seen in Figure – specific muscles, glands and glan.
Response: As previously indicated, despite the fact that we have imaged TAG and 6th ganglia, we have finally decided to eliminate Figure 4 as we could not describe with a high degree of security the structures connected with the nerve roots.
Figure 5
Indicate all structures and regions mentioned in the main text of the MS: epithelium,
ectoperitrophic space, peritrophic membrane, endoperitrophic space, chitin fibers, mycelium-like
particles, nuclei-like particles
Response: This point is much appreciated so it has enriched the descriptions made in figure 5. We have included the explanations to the abbreviations in Figure legend 4 as follows: EN PS: Endoperithropic space; EC PS: Ectoperithrophic space;Myc-lp: Mycelium-like particles; A-lp: Ameboid-like cells; Ch= Chitind ducts, and sclerites (lines 265-270).
Figure 7
Indicate all structures and regions be able to differentiate: epithelium, muscles, inner lumen of the
gut with dense material etc. The picture has to be reader-friendly and informative by itself, without
reading the main text.
Response: Again, this point is much appreciated so it has enriched the descriptions made in figure 6. We have included the explanations to the abbreviations in Figure legend 7 as follows: Rp: Rectal pad; Ep: Epithelium; Inl: Inner lumen; MT: Malpighian tubules; M: Muscle layer (lines 296-300).
Supplementary Figure 1
Please indicate all the structures and regions mentioned in the text. Add scale bar in caption.
BTW, there are no arrows mentioned in the caption.
Response: We have also added this new information into the figure. We have included the scale bar length and explanations to the abbreviations in Figure legend 6 as follows:. Rp: Rectal pad; Ep: Epithelium; Inl: Inner lumen; MT: Malpighian tubules; M: Muscle layer .Scale bar=100nm (lines 296-300).
Reviewer 3 Report
In this paper, the Authors provide interesting results on the abdomen of Apis mellifera workers by using two complementary techniques: micro-ct and histology. Micro-ct allowed the three-dimensional visualisation of the internal structures of the abdomen, emphasising their volumetric attributes and highlighting the spatial relationships between the various parts analysed, thus revealing some features of the internal organs reported here for the first time. The resulting 3D models are certainly of high value and are complemented by an interesting histological investigation, based on the use of different staining to reveal differences in the characteristics of the analysed tissues. The fascinating presence of a new cell type called “ventricular telocytes” is also reported. The present work is certainly interesting as the results are well presented and equally well discussed, however, I would like to point out a few minor issues and concerns encountered throughout the manuscript.
Line 76: I am not a native speaker but I have some reservations about the use of the term "careful" to describe micro CT.
Line 79: I would avoid the use of the term "comparative", after all, only one species is analysed in this study, and no comparative analysis is made.
Lines 80-81: Although there are several reviews on the subject, I think it would be worth expanding this section, adding further clarifying information on the reconstruction of virtual slices and how these are obtained from the original two-dimensional X-ray images derived from the scanning of the rotating sample.
Lines 129-130: the text seems unclear to me, perhaps something is missing after “in 100%”?
Line 135: it is not clear to me what is meant by "brass sample", I assume it is the sample holder. Since it is specified the importance of the alignment of the tube and the pipette tip along the long axis, it might be appropriate to better illustrate how the acquisition setup is designed.
Line 138: Remove “(“ before Xradia.
Lines 153 and 165: I think that the references inserted are incorrect and are not really [1] and [2].
Line 194: I suspect that the use of the term '2d scans' may create ambiguity, suggesting a reference to the raw X-ray images of the specimen during its rotation, rather than the reconstructed virtual sections. Furthermore, I have some doubts about the use of the term 'sclerotic'.
Line 195: Similar to the above, I am afraid that “segmented” could be misleading and referring to the segmentation process used to highlight different structures during the creation of 3D models. Also, are the hairs really segmented?
Lines 209-212: The sentence is unclear to me.
Line 300: Most likely I have missed something, but I wonder why “supplementary figure 1” cannot simply be “figure 8”.
Lines 323 and 326: there is something wrong with the citations, I think [28] should be [27] and that [27] corresponds to [26], I suggest checking them again along the text.
Author Response
In this paper, the Authors provide interesting results on the abdomen of Apis mellifera workers by using two complementary techniques: micro-ct and histology. Micro-ct allowed the three-dimensional visualisation of the internal structures of the abdomen, emphasising their volumetric attributes and highlighting the spatial relationships between the various parts analysed, thus revealing some features of the internal organs reported here for the first time. The resulting 3D models are certainly of high value and are complemented by an interesting histological investigation, based on the use of different staining to reveal differences in the characteristics of the analysed tissues. The fascinating presence of a new cell type called “ventricular telocytes” is also reported. The present work is certainly interesting as the results are well presented and equally well discussed, however, I would like to point out a few minor issues and concerns encountered throughout the manuscript.
Line 76: I am not a native speaker but I have some reservations about the use of the term "careful" to describe micro CT.
Response: We appreciate the referee´ s observation and decided to delete the term “careful” from the sentence”, so the sentence would be as follows: “...requires an accurate methodology that enables…” (lines 76 and 77).
Line 79: I would avoid the use of the term "comparative", after all, only one species is analysed in this study, and no comparative analysis is made.
Response: We really appreciate the observation and we deleted the term “comparative” and added “morphological” so the sentence would be as follows:”...used to analyze the morphological and functional aspects of small animals” (line 80).
Lines 80-81: Although there are several reviews on the subject, I think it would be worth expanding this section, adding further clarifying information on the reconstruction of virtual slices and how these are obtained from the original two-dimensional X-ray images derived from the scanning of the rotating sample.
Response: Thank you very much for the appreciation. We have added the following information in the introduction section that will enrich the background on Micro-CT:
“Micro-CT mainly consists of acquiring images in the path of an x-ray beam [11]. The source is provided by a microfocus x-ray tube that emits x-rays which are collimated and filtered to narrow the energy spectrum. The beam then passes the sample rotated around its long axis and is further detected with a CCD camera. The two-dimensional (2D) trans-axial projections images are collected after sample rotation at different angles.Finally the collection of all 2D images are used for 3D stacking using specialized software [11]. During several years the characterization of new species was performed through manual techniques such as dissection and production of schemes or drawing. This is a non-invasive non-destructive technique that allows for visualization of samples in its original state.” (lines 80-87).
Lines 129-130: the text seems unclear to me, perhaps something is missing after “in 100%”?
Response: We apologize for the mistake, we forgot to add “ethanol” to 100%, so the sentence now stands as follows: “...ethanolic solution of iodine (1%) in 100% Ethanol…” (line 134).
Line 135: it is not clear to me what is meant by "brass sample", I assume it is the sample holder. Since it is specified the importance of the alignment of the tube and the pipette tip along the long axis, it might be appropriate to better illustrate how the acquisition setup is designed.
Response: we have fortunately made a photo of the setup of the holder where the samples are placed in the Micro-CT machine. We generated supplementary figure 2 where the setup is illustrated.
Line 138: Remove “(“ before Xradia.
Response: Done.
Lines 153 and 165: I think that the references inserted are incorrect and are not really [1] and [2].
Response: We really appreciate these observations and apologize for the confusion. The reference [1] was deleted since it was placed there by mistake and the reference [2] was corrected by reference [24] since this is a protocol standardized by one of the authors of the paper (Dr. Kevin Doello).
Line 194: I suspect that the use of the term '2d scans' may create ambiguity, suggesting a reference to the raw X-ray images of the specimen during its rotation, rather than the reconstructed virtual sections. Furthermore, I have some doubts about the use of the term 'sclerotic'.
Response: This point is much appreciated and one which we did not communicate clearly enough in the original text. However, and following referee´s suggestions we have added a new sentence explaining the procedence of the 2D images: “The two-dimensional (2D) trans-axial projections images are collected after sample rotation at different angles” (lines 84-87), thus the picture of Fig.2E is one of this multiple 2D stacks that could be taken alone or used for volumetric 3D reconstruction of the insect. Therefore we decided to indicate in the text that Fig. 2E is an example of scan among all the collected (>1000), Thus the sentence now stands as its follows: “an example of 2D picture collected (Figure 2E) shows the cardiac-valve with its chitinous cuticular surface and some segmented hair emerging on the surface towards the crop”.
Regarding the term sclerotic we apologize for the typo mistake. We have eliminated sclerotic and changed by chitinous: “an example of 2D picture (Figure 2E) shows the cardiac-valve with its chitinous cuticular surface and some segmented hair emerging on the surface towards the crop” (lines 208-210).
Line 195: Similar to the above, I am afraid that “segmented” could be misleading and referring to the segmentation process used to highlight different structures during the creation of 3D models. Also, are the hairs really segmented?
We totally agree with the referee. We decided to eliminate the term “segmented” and indicated as hair-like structures what is shown in the figure. The sentence now stands as follows: “An example of 2D picture collected (Figure 2E) shows the cardiac-valve with its chitinous cuticular surface and some segmented hair-like structures emerging on the surface towards the crop.” (lines 208-210). Additionally, we indicated the hair-like structures in figure 2E.
Lines 209-212: The sentence is unclear to me.
Response: we rewrote the sentence as follows: “The pyloric valve is situated in the connection of the Malpighian tubules and internalized deep into the ileum as shown in Figure 3D-E, Video 1 (Second 1 0:30 to 0:34) and Vídeo 2 (seconds 0:14 to 0:23)” (lines 224-226).
Line 300: Most likely I have missed something, but I wonder why “supplementary figure 1” cannot simply be “figure 8”.
Response: Thank you for the observation, the mistake has been corrected is was supplementary figure 2. (since now we have two supplementary figures).
Lines 323 and 326: there is something wrong with the citations, I think [28] should be [27] and that [27] corresponds to [26], I suggest checking them again along the text.
Response: We appreciate the observation and corrected the citations in the text.
Reviewer 4 Report
The authors present an interesting approach to honey bee anatomy, using novel techniques. These techniques will be very useful for further research. I would like a little more information on the methods, and minor corrections in grammar:
lines161 and 297: is L-S stain the "lactophenol stain" mentioned on line 100?
Sample preparation (lines 127-132 and 150-152) is not completely clear. Was formaldehyde actually a 10% formaldehyde in buffer (as is common)? For how long? Lines 127-132 conflict with lines 150-152. How many minutes in each solvent? How many minutes in melted paraffin and at what temperature? What type of paraffin was used and from what vendor? (Several types of paraffin preparations are made for histological work.)
Line 44 should read "analyses have allowed"
Line 171 should read "composed of 5 ganglia"
Line 329 should read "this work has found"
Line 372 should read "These results are supported"
Line 378 should read "peritrophic matrix, observed after"
Line 382 should read "There are no previous reports"
Author Response
The authors present an interesting approach to honey bee anatomy, using novel techniques. These techniques will be very useful for further research. I would like a little more information on the methods, and minor corrections in grammar:
Response: We really appreciate all the suggestions and the time expended by the referee to enrich and improve the quality of the manuscript. Here above the answers to the comments:
Lines161 and 297: is L-S stain the "lactophenol stain" mentioned on line 100?
Response: We apologize for the confusion. The answer is no. On the line 161 and 297 is explaining Lactophenol-Safranin. We changed “L-S” by “Lactophenol-Safranin”.
Sample preparation (lines 127-132 and 150-152) is not completely clear. Was formaldehyde actually a 10% formaldehyde in buffer (as is common)? For how long? Lines 127-132 conflict with lines 150-152. How many minutes in each solvent? How many minutes in melted paraffin and at what temperature? What type of paraffin was used and from what vendor? (Several types of paraffin preparations are made for histological work.)
Response: In order to clarify, we have remodelled section 2.3. “Tissue staining for optical microscopy” was written again and now states as it follows:
“Formaldehyde fixed samples (10% formalin buffer during 48h) of gut honeybee were included in paraffin after submerging them in consequent increasing ethanol solutions (70%, 90% and 2x absolute ethanol , 1h each), toluene (3x1h,), melted paraffin (60ºC during 3x3h, Sigma-Aldrich), finally mounting paraffin embedded blocks. Staining with PAS-AB (Merck) was performed to deparaffinized (xylene 2x10min) and hydrated (subsequent decreasing concentrated ethanol solutions absolute, 90% and 70%, 2x10min each) 0.5μm sections[1]. All samples were stained using different staining protocols. The first one, included incubation of the samples for 15 min in Alcian Blue, wash in tap water for 2 minutes and then in distilled water once more. Next, a 5-min incubation with Periodic Acid (in darkness) was performed followed by a single wash with tap water. So far, all steps were performed at RT. Then the samples were stained with Shiff acid in darkness and incubated for 15 min on ice. Next, samples were washed with tap water for 5 min and stained with hematoxylin for 1 min. Finally the samples were washed again in tap water for 2 minutes, dehydrated in increasing ethanol solutions and xylene, following standard protocol, and finally mounted using Organo/Limonene Mount™(Sigma). The second staining protocol (L-S staining) included the incubation of the samples in Lactophenol Blue, (RT, 20 min), wash with tap water (1 min) and then staining with Safranin O (10sec). Then, the samples were washed with tap water and left to dry at RT. Finally, samples were dehydrated in increasing ethanol solutions and xylene, following standard protocol, and finally mounted using Organo/Limonene Mount™(Sigma). The third staining protocol, Pentachome staining, was performed as previously described by Doello (2014) [2]. Hematoxylin-eosin staining was developed using standard protocol. Briefly, samples were stained for 5 min with hematoxylin and washed in tap water for 10 min. Finally, samples were stained in eosin for 5 min. Finally the samples were washed again in tap water for 2 minutes, dehydrated in increasing ethanol solutions and xylene, following standard protocol, and finally mounted using Organo/Limonene Mount™(Sigma)” (lines 159-184).
Line 44 should read "analyses have allowed"
Response: Done
Line 171 should read "composed of 5 ganglia"
Response: We change the sentence by “Abdomen contains several components of digestive, circulatory, respiratory, nervous and genital systems” (lines 187 and 188).
Line 329 should read "this work has found"
Response: Done
Line 372 should read "These results are supported"
Response: Done
Line 378 should read "peritrophic matrix, observed after"
Response: Done
Line 382 should read "There are no previous reports"
Response: Done
This manuscript is a resubmission of an earlier submission. The following is a list of the peer review reports and author responses from that submission.
Round 1
Reviewer 1 Report
Dear authors,
First Apis mellifera is native in Africa and Europe and therefore the common name “Western honey bee” should be used, especially since in Spain you have a strong influence of the A-lineage it is even more wrong to speak of European honey bees.
I miss any information on samples sizes and any information in how many samples one finds these new structure - Is it only anecdotal because you have only one sample ?
Also were the staining and CT done on the same samples of different ones, I assume different ones. But then please state that this is the case.
Also, state the subspecies you used.
The literature is rather selectively reviewed, I miss the classic anatomy work by Snodgrass among others.
Also, it has been described before that there is a connection
To our knowledge, this is the first report describing the connection between the 322 sting apparatus and nervous system.
Moreover, what do you think is the function ?
The ms is full of formatting and spelling mistakes, species are not in italics, esp in the reference list.
The discussion should take it beyond the description and explained or speculate about the functions. Moreover, the conclusion is a brief summary rather than a conclusion. Overall, I miss the novel aspects and find it very descriptive. What is different to Smith et al 2006?
Reviewer 2 Report
The manuscript describes some unrelated methods of imaging honey bee anatomy. It is not clear what is novelty of the manuscript. The use of advanced imaging is not novel in case of honey bee. Some important references were not included:
Tomanek, B., Jasiński, A., Sułek, Z., Muszyńska, J., Kulinowski, P., Kwieciński, S., ... & Kibiński, J. (1996). Magnetic resonance microscopy of internal structure of drone and queen honey bees. Journal of Apicultural Research, 35(1), 3-9.
There is even a review relate to this:
Hart A.G., Bowtell R.W., Köckenberger W., Wenseleers T., Ratnieks F.L.W. (2003). Magnetic resonance imaging in entomology: a critical review. Journal of Insect Science, 3(1).
Many of the claims of novelty are not correct. For example in line 322 there is claim: "To our knowledge, this is the first report describing the connection between the sting apparatus and nervous system."
At least one publication describing in details innervations of sting apparatus was omitted:
Ogawa, H., Kawakami, Z., & Yamaguchi, T. (1995). Motor pattern of the stinging response in the honeybee Apis mellifera. The Journal of experimental biology, 198(1), 39-47.
There are many important errors. Already in the first sentence (line 24) there is information that Apis mellifera belongs to "hemiptera phylum". The rank of Hemiptera is not phylum but order. Apis mellifera belongs to Hymenoptera.
It is possible that there is some new information about telocytes in ventriculus, however, most of the manuscript is not suitable for publication.